# Body Donation Registration in Taiwan: Reasons and Associated Psychological Factors

**DOI:** 10.3390/healthcare11070969

**Published:** 2023-03-28

**Authors:** Wan-Lan Chen

**Affiliations:** Department of Human Development and Psychology, Tzu Chi University, Hualien 970048, Taiwan; wanlanchen@mail.tcu.edu.tw

**Keywords:** altruism, body donation, death anxiety, gratitude, purpose in life

## Abstract

Dissection is an essential element of medical training and depends on the availability of cadavers. However, traditional Chinese culture widely regards the body as a gift from one’s parents that should remain intact after death, resulting in a shortage of cadavers for medical training and research. This situation changed in Taiwan when Master Cheng Yen, the founder of the Buddhist Tzu Chi Medical Foundation, supported the donation of bodies to medical science. This study aimed to investigate the current situation of body donation in Taiwan, including donors’ motivation and psychological characteristics. A questionnaire was conducted with 681 adult participants, including 336 people who pledged to donate their bodies to medical science after death and a control group comparable in age, gender, and level of education. All participants answered questions regarding anxiety over death, purpose in life, gratitude, altruism, and life satisfaction. In addition, the registered donor group answered questions regarding the motivation for donating their bodies to science. The main influencing factors were to help advance medical science, make a positive contribution to society, and release attachment from the body. Further, many male participants indicated the desire to reduce trouble and expenses that their families would incur in making funeral and burial or cremation arrangements. The main predictors of donating one’s body to medical science were low anxiety concerning death, a high level of altruism, and gratitude.

## 1. Introduction

Dissection and simulated surgery are essential to medical training and medical science [1,2]. Therefore, access to a sufficient number of cadavers is essential [3]. Before the mid-1990s, most of the cadavers used in medical schools in Taiwan were anonymous, and only a handful of people chose to donate their body for such use [4]. In Chinese societies, there are various death rituals which can be sources of comfort for families at funeral and burial proceeding; however, the traditional rituals cannot be carried out if one donated his/her body to anatomy institutes. This may be one of the reasons for the shortage of cadavers in Taiwan [5]. In addition, Confucian societies attach importance to maintaining the integrity of the body in both life and death [6]. In the traditional Confucian value system, which continues to be highly influential in Taiwan [7], one’s body is regarded as a gift from one’s parents, which should be kept intact as an expression of filial piety in both life and death [8]. As such, in traditional Chinese societies, an auspicious ending of a human life through interment or cremation requires an intact body [9].

In 1994, the Buddhist Tzu Chi Foundation, which is highly influential in Taiwan and among Chinese people overseas [10], established the Tzu Chi College of Medicine (currently Tzu Chi University [TCU]). The founder, Master Cheng Yen, worked to promote grassroots philanthropy and hands-on social service [5] and encouraged people to donate their bodies to medical science to benefit society [11]. TCU started its first gross anatomy course in 1996 and used 1051 corpses for dissection or simulated surgery in the TCU Medical Simulation Center (MSC-TCU) as of 2021 [12]. The bodies were pledged by individuals prior to their death, a unique phenomenon among medical schools in Taiwan [13].

Since Master Cheng Yen advocated that the donation of body to medical science was an act of altruism, gratitude, and love, she cautioned against regarding cadavers as merely instructional materials and argued that they should be respected as teachers [10]. Therefore, the cadavers used at MSC-TCU are referred to as ‘silent mentors’ [5], a practice that has been adopted in other medical schools in Taiwan [14] and in other regions [15]. For all dissections and simulated surgeries conducted at MSC-TCU, the participating students and doctors attend a memorial service acknowledging the donor’s generosity, during which prayers are recited and Buddhist scriptures are chanted on behalf of the donor, whose family is invited to attend [4,16]. In contrast, in most other countries, cadavers remain anonymous [17]. Every year, 9 to 11 memorial services are held at Tzu Chi University, and widespread media coverage has resulted in a significant increase in the donation of bodies to medical science [13]. The cumulative number of people who pledged to donate their bodies to MSC-TCU was 387 in 1995, 9902 in 2005, and over 40,000 in 2021 [12].

Although the number of donors has grown substantially, research regarding the motivation to donate one’s body to science is scarce. The theory of planned behavior states that health-related decisions are based on rational considerations [18], with influencing factors such as knowledge and attitude, intentions of the donor and their loved ones [19,20,21], and prevailing social norms [22,23,24]. However, research suggests that the decision to donate one’s organs or body to medical science is beyond merely rational [25,26] and is influenced by psychological factors, such as altruism [27], gratitude [2], and spiritual factors, such as purpose in life and anxiety regarding death [28].

To date, research on the psychological factors related to body donation has mostly been limited to Western countries. To initiate an exploratory study on the increasing number of registered body donors, this study aimed to elucidate the factors influencing the decision to donate one’s body to medical science. This study examined anxiety concerning death, purpose in life, gratitude, altruism, and life satisfaction among donors and non-donors. Furthermore, this study sought to identify factors that had the strongest predictive value for donating one’s body to medical science.

## 2. Materials and Methods

### 2.1. Procedures and Participants

This study was conducted between February 2018 and January 2021. The participants were recruited by convenience sampling in several Jing Si Halls in northern and eastern Taiwan, where volunteers of the Tzu Chi Foundation and local residents usually gather for community events. Five trained researchers issued face-to-face invitations to participate in the study. Informed written consent was obtained from all participants prior to participation. Participants were given vouchers worth an equivalent of USD 7 upon completion of self-reported measures. The study was approved by the Research Ethics Committee of Hualien Tzu Chi Hospital, Buddhist Tzu Chi Medical Foundation (IRB106-175-A).

A total of 681 participants with a mean age of 58.24 (SD = 10.56) were recruited for the study, including 336 people who registered for body donation and 345 controls, who were matched for gender, age, and level of education with the donor participants. The inclusion criteria were being between 25 and 75 years old and having no current diagnoses of psychiatric disorders. The demographic characteristics of the two groups are presented in Table 1. Additionally, the mean age for registering to be a body donor was 48.51 years (SD = 10.47) for women and 47.39 years (SD = 10.13) for men.

### 2.2. Measures

#### 2.2.1. Reasons for Body Donation Registration

A modified version of motivations for donation [2] was utilized to assess the reasons for body donation among the donor group using binary responses (0 = no; 1 = yes). A sample item is “I want to contribute to medical science”.

#### 2.2.2. Life Satisfaction

The Satisfaction With Life Scale (SWLS), developed by Diener et al. [29], is a self-reporting measure to assess global life satisfaction, consisting of 5 items rated on a 7-point Likert scale (1 = strongly disagree, 7 = strongly agree). Despite consisting of only five items, the SWLS has consistently demonstrated good psychometric properties. This has contributed to its widespread use in research and clinical settings as a reliable and valid tool for assessing an individual’s subjective well-being [30]. The original SWLS demonstrated a high internal consistency (ranging from 0.79 to 0.89) and good test-retest correlations (0.84). The Chinese version of the SWLS [31] was utilized for this study. The internal consistency of the SWLS in the present study was 0.85.

#### 2.2.3. Altruism

A modified version of the prosocial attitude and behavioral intention test [32] was used to assess altruism. For this assessment, participants read a short newspaper article on a typhoon-induced disaster that occurred in 2016 in the eastern region of Taiwan. The measure consisted of 5 items rated on a 6-point Likert scale (1 = strongly disagree, 6 = strongly agree). The sample items are: “The Taiwanese government has the responsibility to help the inhabitants of the region affected by the typhoon” to measure altruistic attitude, and “I feel a personal obligation to donate money to help victims of the typhoon” to measure altruistic behavior. The internal consistency of the measure was 0.88 in the present study.

#### 2.2.4. Purpose in Life

Purpose in life was assessed using a five-item subscale from the meaning in life questionnaire (MLQ), developed by Steger et al. [33]. A sample item is “I understand my life’s meaning.” The answers were rated on a 7-point Likert scale (1 = completely disagree, 7 = completely agree). The MLQ subscale was translated using a three-procedure approach [34], where the scale was translated into Chinese, the Chinese version was translated back to English, the equivalence between the two versions was evaluated, and the Chinese questionnaire was revised as required. The internal consistency of subscale was 0.82 in the present study.

#### 2.2.5. Gratitude

Gratitude was measured using the Gratitude Questionnaire (GQ) developed by McCullough et al. [35], which is a six-item self-reported measure that assesses dispositional gratitude. A sample item is “I have a lot in life to be thankful for.” The responses were rated on a 7-point Likert scale (1 = strongly disagree, 7 = strongly agree). The Chinese version of the GQ [36] was used in this study. The alpha coefficient was 0.80 in Chen et al. [36] (2009) and 0.68 in the present study.

#### 2.2.6. Death Anxiety

The Templer Death Anxiety Scale (TDAS) developed by Royal and Elahi [37] is a 15-item scale used to measure the level of death anxiety. A sample item is “I fear dying a painful death,” rated on a 5-point Likert scale (1 = completely disagree, 5 = completely agree), with higher scores indicating greater death anxiety. The Chinese version of the TDAS (CL-TDAS) developed by Yang and her colleagues [38] was used in this study. The alpha coefficient of the CL-TADS was 0.82 [38]. The internal consistency of the scale in the present study was high (Cronbach’s α = 0.81).

### 2.3. Data Analysis

To compare the differences in characteristics between the registered donor and the non-registered control groups, statistical analyses were conducted using χ^2^ tests for categorical variables and independent *t*-tests for continuous variables. Then, nonparametric univariate analyses were performed using the Mann-Whitney U test to identify differences in psychological factors between the donor and control groups. The factors with a *p* value < 0.05 were chosen for univariate as well as multivariate logistic regression analyses to identify the factors influencing body donation. All analyses were conducted using IBM SPSS statistics for Windows, version 22.0 (IBM Corp., Armonk, NY, USA).

## 3. Results

Prior to examining the factors influencing the decision to body donation, a series of chi-square tests were performed to compare the demographic variables of the registered donor and the control groups. No significant differences were observed between the 2 groups in terms of demographic characteristics, including gender (χ^2^ (*df* = 1) = 0.06, *p* = 0.81), level of education (χ^2^ (*df* = 4) = 1.70, *p* = 0.79), employment status (χ^2^ (*df* = 1) = 1.31, *p* = 0.25), and marital status (χ^2^ (*df* = 2) =0.99, *p* = 0.61). An independent *t*-test was performed to test whether significant regional differences (northern versus eastern Taiwan) existed on the research variables. The results revealed no significant differences on the scores of death anxiety (*t*(679) = −0.25, *p* = 0.81), gratitude (*t*(679) = 1.15, *p* = 0.25), altruism (*t*(679) = 0.32, *p* = 0.75), life meaning (*t*(679) = 0.40, *p* = 0.69), and life satisfaction (*t*(679) = 1.04, *p* = 0.30).

Figure 1 shows the differences in the reasons for body donation by gender. Nearly 90% of the donors indicated over two reasons for donation. The most common reasons were full utilization of the body after death (94.0%), advancing medical education (91.1%), contributing to medical science (86.6%), and being useful to society (80.1%). In addition, more than half of the participants indicated releasing attachment from their bodies as a reason for donation. Significantly more male than female donors indicated that they wished to avoid burdening their families with funeral arrangements.

Table 2 presents the zero-order correlations between the study variables for the two groups. In the registered donor group, all variables were negatively associated with the levels of death anxiety. Similarly, all variables except altruism and gratitude were significantly related to death anxiety in the non-registered control. The correlation between altruism and death anxiety was significantly higher for the donor group than the control group (*z =* 2.28, *p <* 0.05). In both groups, all variables except death anxiety were significantly related in the positive direction. The correlations between altruism and gratitude and gratitude and purpose in life were significantly higher for the donor group than the control group (*z =* 2.99, *p <* 0.01 and *z =* 2.49, *p <* 0.05, respectively).

The Mann-Whitney U test revealed significant differences between the donor and control groups in all psychological factors. Table 3 presents the means and standard deviations of the factors for the donor and control groups. The donors had significantly higher scores for altruism, gratitude, purpose in life, and life satisfaction, and lower scores for anxiety over death.

Next, logistic regression models were utilized to estimate the odds ratios of body donation for psychological factors. Table 4 presents the results of the univariate analyses and a multivariate analysis; registration as body donors was the outcome categorical variable, and age, gender, and psychological variables were the predictors. The univariate analyses indicated that all predictors but gender were related to the registration of body donation. However, in the multivariate logistic regression model, significant differences were observed only for anxiety over death, altruism, and gratitude, while purpose in life and life satisfaction were not statistically significant. The highest magnitude was for altruism (*OR* = 1.17, *p* < 0.00), followed by anxiety over death (*OR* = 0.95, *p* < 0.00), and gratitude (*OR* = 1.06, *p* < 0.00). Lower scores for anxiety over death and higher scores for altruism and gratitude were significant contributing factors in body donation. These 3 factors explained 31% of the difference between the donor and control groups.

## 4. Discussion

The aim of this present study was to investigate the psychological characteristics and motivations of individuals who registered as body donors for medical science in Taiwan. The results of this study revealed that the primary motivation for body donation was the desire to be useful to society after death and to advance medical education and research. Interestingly, this study also found that, although women were more likely to pledge to donate their bodies, the majority of donated bodies were male. This study further identified the key factors that predict pledging to donate one’s body to medical science. The results showed that altruism was the most important factor in predicting donation, and that registered donors exhibited higher levels of altruism compared to the control group. Additionally, donors had lower levels of anxiety related to death and stronger feelings of gratitude, purpose in life, and life satisfaction compared to non-donors. However, the study found that only death anxiety and gratitude significantly predicted an individual’s likelihood of donating their body to medical science.

The number of women registered as body donors was twice that of men in this sample. According to the website of MSC-TCU, among the 40,000 people who pledged to donate their bodies to the MSC-TCU, approximately 62% were women. Similarly, approximately 66% of the participants in this study were women. This is in line with previous studies on whole-body donations [24,39,40], which may be due to the higher propensity of women to participate in religious, spiritual, and civic activities, which tend to stress altruism [40]. The influence of this factor requires further research. Interestingly, most of the bodies actually donated to MSC-TCU (*n* = 1049) were male (59%). Body donation to the MSC-TCU has several conditions, and reasons for rejection include major reconstructive surgery, large unhealed wounds, edema caused by pharmaceuticals or disease, obesity, and emaciation [12]. The average life expectancy in Taiwan is 84.7 years for women and 78.1 years for men [41]; therefore, women’s bodies may be less likely to meet the acceptance criteria. This also requires additional research.

Most of the donors surveyed in the present study indicated that they wanted to help advance medical education and research, which is in line with the previous research [2,39,42,43,44,45,46,47,48]. A remarkable consistency in motivational factors for body donation has been observed across various studies worldwide, which includes desires to contribute to medical education or scientific research or be useful or helpful. Farsides et al. [45] classified these motivations as “medical altruism”, referring to individuals who seek to benefit healthcare professionals or institutions. In addition, the desire to spare one’s family the trouble and expense related to one’s funeral was one of the main motivations among the male participants in this study, but not the female participants. Farsides et al. [45] labeled the motivation as “intimate altruism”, where some individuals choose to forego a traditional funeral, at least in part, with the aim of preventing their loved ones from bearing the cost, grief, or inconvenience associated with it. In Chinese societies such as Taiwan, male relatives are expected to play a more prominent role in the funerals of family members [5]; therefore, it is likely that their past experiences made male donors more aware of and sensitive to issues related to arranging funerals and interment or cremation.

In the study, over 50% of the individuals who registered as body donors expressed the desire to release attachment from the body, suggesting a gradual shift away from the traditional emphasis in Chinese societies on keeping the corpse intact. In Taiwan, the traditional Confucian value places great importance on filial piety and emphasizes that one’s body is a precious gift from one’s parents that must be respected and kept intact throughout one’s life and even after death [8]. However, Master Cheng Yen teaches that donating one’s body to medical science is a commendable act of filial piety, as one’s physical form becomes useless to oneself after death, so it can still be beneficial to others [5].

Donors had significantly higher levels of altruism compared to the control group, and altruism was found to have the highest predictive value for donating one’s body to medical science, which is in line with previous studies [27]. Altruism is generally understood as a voluntary and deliberate action that benefits others, is anonymous, and is free of selfish motivation [49]. The donors in the present study were self-motivated, as they all submitted applications to MSC-TCU for which they received no compensation. However, as their names and backgrounds are mentioned prior to the autopsy and in the memorial ceremony held at the MSC-TCU, the donors do not remain anonymous. Nevertheless, this occurs after the donor’s death, with no interaction between the donor and beneficiaries. Bar-Tal [50] argued that one of the characteristics of altruism is that the actor may have to pay a certain price, which in the case of whole-body donation can include negative reactions from one’s family when they learn about their non-traditional decision [5]. Bolt et al. [2] stated that engaging in a virtuous act, such as donating one’s body to medical science, can enhance the donor’s positive self-image, and that making a definitive decision regarding one’s body after death may increase one’s sense of self-efficacy. In line with this, we observed that many donors in this study were motivated by a desire to reduce their attachment to the body, which can be seen as a form of self-benefit.

Compared to the control group, the donors had significantly lower levels of anxiety over death and a stronger sense of purpose in life; however, only anxiety over death significantly predicted an individual’s likelihood of donating one’s body to medical science. Previous studies found a positive correlation between low anxiety over death and donating one’s body to medical science [28,51,52,53,54]. An individual’s beliefs regarding death are largely influenced by their cultural background [55]. In Chinese culture, direct references to death are regarded as inauspicious [56,57], and there are numerous euphemisms for death, such as “returning home” and “a withered leaf falling to the root” [58]. Although such euphemisms may help relieve the sadness and fear associated with death, they are also a way of avoiding the inevitability of death [59]. It may be argued that pledging one’s body to medical science is a way of confronting death; however, some see it as inviting misfortune [25]. Such concerns may be a key factor dissuading people from body donation. Hirschberger et al. [60] found that participants who were asked to reflect on their own death were less likely to sign an organ donation consent form. Similarly, in the present study, a positive correlation was found between low anxiety over death and the likelihood of donating one’s body to medical science. However, the reverse may also be true. The decision to donate one’s body could result in lower anxiety regarding death, as it may provide a clearer picture of what will happen to one’s body after death and what kind of funeral ceremony will be performed [5].

Donors and non-donors in the present study demonstrated significant differences in the measures of gratitude and life satisfaction. However, only a high sense of gratitude was found to be a strong predictor of donating one’s body to medical science. Similarly, McCullough et al. [61] found that a relatively high sense of gratitude, whether self- or other-rated, was a reliable predictor of prosocial behaviors. The sense of gratitude is a type of ethical sentiment associated with concern for the well-being of others, predisposing one to altruistic behavior [62]. Bolt et al. [63] found that a large proportion of people who donated their bodies to medical science were motivated by the desire to express gratitude for medical care received in the past or the benefits brought to society in general by advancements in medical science. Some registered donors may express gratitude for the positive impact that medical science has had on the health and well-being of their family. When a person receives successful treatment, their family often feels a sense of relief and gratitude as well. This shared gratitude can inspire a desire to give back to medical science in some way.

Research on motivations for donating one’s body to medial science and associated psychological factors in Taiwan is scarce. This study advanced the understanding of this topic; however, it had some limitations. First, due to the cross-sectional methodology used to collect data, the causal relationship between donation and the variables measured in this study remains somewhat conjectural; that is, while the correlation is clear, the causation is not. In addition, the questionnaire used in this study covered a limited set of mainly positive factors influencing the decision to donate one’s body to medical science; however, it is likely that other factors are influential. Therefore, future studies should utilize open-ended questions that would allow a wider range of motivations to be included in the data. Finally, as data were self-reported, some responses may have been influenced by social expectations; that is, some responses may have been motivated by the desire to project a positive image and serve as a role model for others rather than true views and feelings. Therefore, future research should utilize more objective measures, such as responses provided by acquaintances and other third parties.

## 5. Conclusions

The act of body donation provides a crucial resource for medical education and research. Nonetheless, there is limited knowledge about the psychological characteristics of individuals who register for body donation. This study identified the factors that predict pledging to donate one’s body to medical science. The results found that altruism was the most important factor in predicting donation, and that registered donors had higher levels of altruism compared to the control group. Donors also had lower levels of anxiety over death and stronger senses of gratitude, purpose in life, and life satisfaction compared to non-donors. However, only death anxiety and gratitude significantly predicted an individual’s likelihood of donating one’s body to medical science. The results showed that the desire to be useful to the society after death and help advance medical education and research were the primary motivations behind body donation. The study also noted that women were more likely to pledge to donate their bodies, but that most bodies actually donated were male. Further research is needed to investigate the reasons for this discrepancy, including whether women’ bodies may be less likely to meet the acceptance criteria for donation by the time they are deceased.

## Figures and Tables

**Figure 1 healthcare-11-00969-f001:**
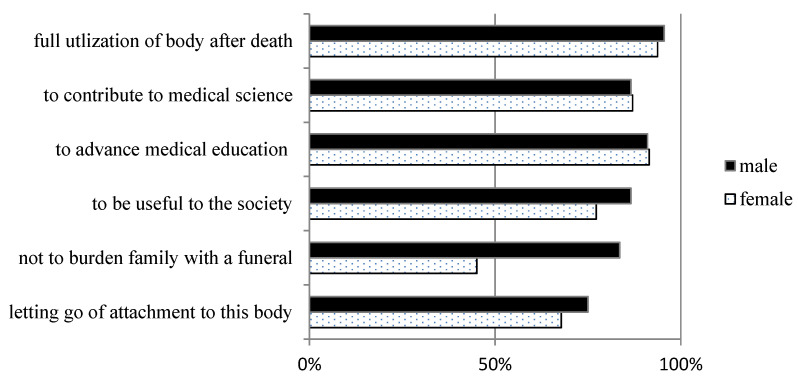
Reasons for body donation among male and female registered donors. Total number of respondents = 336.

**Table 1 healthcare-11-00969-t001:** Demographic characteristics of the two groups.

	Registered Donor (*n* = 336)		Non-Registered Control (*n* = 345)	
	*n*	%	*n*	%
Gender				
female	224	66.67	227	65.80
male	112	33.33	118	34.20
Age (years)				
50	61	18.15	64	18.55
51–65	190	56.55	201	58.26
66+	85	25.30	80	23.19
Education				
middle school or less	94	27.98	88	25.51
high school	115	34.23	116	33.62
2-year college to college degree	107	31.84	124	35.94
graduate or professional school	20	5.95	17	4.93
Employment status				
full or part-time employee	190	56.55	209	60.58
retired, unemployed, or student	146	43.45	136	39.42
Marital status				
married or living with partner	228	67.86	246	71.30
separated, divorced, or widowed	74	22.02	69	20.00
never married	34	10.12	30	8.70

**Table 2 healthcare-11-00969-t002:** Zero-order correlations among study variables.

	1	2	3	4	5
Death anxiety		−0.09	−0.09	−0.26 **	−0.26 **
2. Altruism	−0.26 *		0.39 **	0.32 **	0.26 *
3. Gratitude	−0.11 *	0.18 **		0.48 **	0.22 **
4. Purpose in life	−0.28 **	0.25 **	0.32 **		0.46 **
5. Life satisfaction	−0.18 **	0.18 **	0.18 **	0.44 **	

Correlations above the diagonal are for the non-registered control (*n* = 345); correlations below the diagonal are for the registered donors (*n* = 336). * *p <* 0.05. ** *p <* 0.01.

**Table 3 healthcare-11-00969-t003:** Means and standard deviations of research variables for registered donor and non-registered control groups.

	Registered Donor (*n* = 336)	Non-Registered Control (*n* = 345)	Mann-Whitney Z Scores
	M (SD)	M (SD)	
Death anxiety	35.59 (8.82)	42.19 (9.85)	−8.60 ***
Altruism	26.00 (3.13)	22.33 (5.04)	−10.34 ***
Gratitude	35.68 (4.48)	32.70 (5.33)	−7.38 ***
Purpose in life	31.51 (4.08)	28.53 (5.28)	−8.04 ***
Life satisfaction	28.19 (4.86)	26.23 (5.09)	−5.01 ***

*** *p* < 0.001.

**Table 4 healthcare-11-00969-t004:** Univariate and multivariate logistic regression model summary.

Variables	Wald χ^2^	Crude OR	95% CI		Wald χ^2^	Adjusted OR	95% CI	
			LL	UL			LL	UL
Age	4.40 *	1.02	1.00	1.03	0.57	1.01	0.99	1.03
Gender	0.06	1.4	0.76	1.43	0.02	0.97	0.67	1.42
Death anxiety	67.85 ***	0.93	0.91	0.94	29.47 ***	0.95	0.93	0.97
Altruism	91.69 ***	1.25	1.20	1.31	43.66 ***	1.17	1.12	1.23
Gratitude	53.48 ***	1.06	1.02	1.17	8.08 **	1.06	1.02	1.17
Purpose in life	91.69 ***	1.25	1.20	1.31	2.41	1.03	0.99	1.09
Life satisfaction	22.29 ***	1.08	1.04	1.11	0.05	0.99	0.96	1.03

OR, odds ratio; CI, confidence interval. * *p* < 0.05. ** *p <* 0.01. *** *p <* 0.01.

## Data Availability

Relevant data supporting reported results can be made available from corresponding author on request.

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
