# Peer review of "Body Donation Registration in Taiwan: Reasons and Associated Psychological Factors"

_healthcare, 2023, doi:10.3390/healthcare11070969_

Round 1

Reviewer 1 Report

Dear Authors;

I would like to review it again after extensive statistical analysis is done. Best regards

Author Response

Dear reviewer,

Thank you for your very careful review of this manuscripts, and for the suggestions that ensued. The responses are given as below:

Point 1: I would like to review it again after extensive statistical analysis is done.

Response 1: Thanks for pointing this out. Further analysis has been conducted to enhance the accuracy and depth of the findings.  

2.3 Data analysis

To compare the differences in characteristics between the registered-donor and non-registered control groups, statistical analyses were conducted by using χ2 tests for categorical variables and independent T-tests for continuous variables.(p.4)

  1. Results

Prior to examining the factors influencing the decision to body donation, a series of chi-square tests were performed to compare demographic variables of the registered-donor and the control groups. No significant differences were observed between the two groups in terms of demographic characteristics, including gender (χ2 (df = 1) = 0.06, p = .81), level of education (χ2 (df = 4) = 1.70, p = .79), employment status (χ2 (df = 1) = 1.31, p = .25), and marital status (χ2 (df = 2) =0.99, p = .61). An independent t-test was performed to test whether significant regional difference (northern versus eastern Taiwan) exists on the research variables. The results revealed no significant differences on scores of  death anxiety (t(679) = -.25, p = .81), gratitude (t(679) = 1.15, p = .25), altruism (t(679) = .32, p = .75), life meaning (t(679) = .40, p = .69), and life satisfaction (t(679) = 1.04, p = .30). (p.4)

Table 2 presents zero-order correlations between the study variables for the two groups. In the registered-donor group, all variables were negatively associated with the levels of death anxiety. Similarly, all variables except altruism and gratitude were significantly related to death anxiety in the non-registered control. The correlation between altruism and death anxiety was significantly higher for the donor group than the control (z = 2.28, p < .05). In both groups, all variables except death anxiety were significantly related in the positive direction. The correlations between altruism andgitude, and gratitude and purpose in life were significantly higher for the donor group

than the control (z = 2.99, p < .01 and z = 2.49, p < .05, respectively). (p.5)

Next, logistic regression models were utilized to estimate the odds ratios of body donation for psychological factors. Table 4 presents the results of the univariate analyses and a multivariate analysis, registration as body donors was the outcome categorical variable, and age, gender and psychological variables were predictors. The univariate analyses indicated that all predictors but gender were related to registration of body donation. However in the multivariate logistic regression model, significant differences were observed only for anxiety over death, altruism, and gratitude, while purpose in life and life satisfaction were not statistically significant. (p.5)

Reviewer 2 Report

The manuscript deals with an interesting topic, entitled Body donation in Taiwan. I think a survey on this topic is very useful and therefore the perspective provided by the article is certainly of interest in the field of consent relate to the body donation in specific Countries. The manuscript is well organized, the writing is clear and concise and the methodology is correct and well described.  

This could also be a starting point for other groups of researchers, perhaps from other countries of the world, to be able to see if there are actually cultural factors that can influence the response to public engagement in body donation.

Nevertheless, I have the following concerns:

- The survey is based on the activity of 5 interviewers. It would be useful for the reader to know if the interviews are based on fixed questions, and in this case the authors could attach the text of the questionnaire. Differently, it was only asked the motivation to the donation? In any case it would be useful to deepen this part to better clarify the content.

- The numbers in the columns in table.1 are nonaligned , so it is not possible to interpret the data reported.

- In the discussion ( pag.5 starting from the last 2 lines9 the paragraph that starts with “ The traditional Confucian value system,….” Till the end ( “… funerals and interment or cremation”) is a an explanation more adapted to an introduction rather than a discussion. It is better to move to introduction, eventually in the discussion the Authors can refer to this sentence.

Author Response

Response to Reviewer 2 Comments

Thank you for your very careful review of this manuscripts, and for the suggestions that ensued. The responses are given as below:

Point 1: The survey is based on the activity of 5 interviewers. It would be useful for the reader to know if the interviews are based on fixed questions, and in this case the authors could attach the text of the questionnaire. Differently, it was only asked the motivation to the donation? In any case it would be useful to deepen this part to better clarify the content.

Response 1: Thank you for pointing this out. Indeed, it might be misleading. The data collection process involved participants completing self- reported measures. Research assistants issued face-to-face invitations to participate in the study, they didn’t conduct interview. In the revision, the sentence has been revised as suggested. 

  This study was conducted between February 2018 and January 2021. Participants were recruited by convenient sampling in several Jing Si Halls in northern and eastern Taiwan, where volunteers of the Tzu Chi Foundation and local residents usually gather for community events. Five trained researchers issued face-to-face invitations to participate in the study. Informed written consent was obtained from all participants prior to participation. Participants were given vouchers worth an equivalent of 7 USD upon completion of self-reported measures.

Point 2: The numbers in the columns in table.1 are nonaligned , so it is not possible to interpret the data reported.

Response 2: Thanks for pointing this out. Table 1 has be modified in the revision.

(p.3)

Table 1. Demographic characteristics of the two groups.

Registered donor

(N=336)

Non-registered Control (N=345)

n

%

n

%

  Gender

female

224

66.67

227

65.80

male

112

33.33

118

34.20

  Age (years)

50 -

 61

18.15

 64

18.55

51-65

190

56.55

201

58.26

66+

 85

25.30

 80

23.19

  Education

middle school or less

94

  27.98

88

25.51

high school

115

34.23

116

33.62

     2 year college to college degree

107

31.84

124

35.94

     graduate or professional school

 20

 5.95

 17

 4.93

Employment status

 full or part time employee

190

56.55

209

60.58

     retired, unemployed, or student

146

43.45

136

39.42

Marital status

  married or living with partner

228

67.86

246

71.30

    separated, divorced, or widowed

 74

22.02

 69

20.00

never married

 34

10.12

 30

 8.70

Point 3: In the discussion ( pag.5 starting from the last 2 lines9 the paragraph that starts with “ The traditional Confucian value system,….” Till the end ( “… funerals and interment or cremation”) is a an explanation more adapted to an introduction rather than a discussion. It is better to move to introduction, eventually in the discussion the Authors can refer to this sentence.

Reponse 3: Thanks for the comments. I have revised the relevant section and made the content clearer. 

  1. Introduction

Dissection and simulated surgery are essential to medical training and medical science [1-2]. Therefore, access to a sufficient number of cadavers is essential [3]. Before the mid-1990s, most of the cadavers used in medical schools in Taiwan were anonymous, and only a handful of people chose to donate their body for such use [4]. In Chinese societies there are various death rituals which can be sources of comfort for family at funeral and burial proceeding; however, the traditional rituals cannot be carried if one donated his /her body to anatomy institutes. This may be one of the reasons for the shortage of cadavers in Taiwan [5]. In addition, Confucian societies attach importance to maintaining the integrity of the body in both life and death [6]. The traditional Confucian value system, which continues to be highly influential in Taiwan [7], one’s body is regarded as a gift from one’s parents, which should be kept intact as an expression of filial piety in both life and death [8]. As such, in traditional Chinese societies, an auspicious ending of a human life through interment or cremation requires an intact body [9]. (p.1)

  1. Discussion

In the study, over 50% of the individuals who registered as body donors expressed the desire to release attachment from the body, suggesting a gradual shift away from the traditional emphasis in Chinese societies on keeping the corpse intact. In Taiwan, the traditional Confucian value places great importance on filial piety and emphasizes that one’s body is a precious gift from one’s parents that must be respected and kept intact throughout one’s life and even after death [8]. However, Master Cheng Yen teaches that donating one’s body to medical science as a commendable act of filial piety, as one’s physical form becomes useless to oneself after death, it can still be beneficial to others [5]. (p. 7)

For more details please see the revised version manuscript.

Reviewer 3 Report

1.      The cited literature is too old, it is recommended to add some recent literature.

2.      The research method should be explained in more detail. Why choose to use convenience sampling instead of all the volunteers in Jing Si Halls as the sample group? However, the research subjects come from two completely different living areas in northern Taiwan and eastern Taiwan. Whether the region will cause differences should be included in the control.

3.      Reasons should be given for choosing the scale to be used. For example, there are many quality of life scales. Why choose the SWLS developed in 1985? The development of SWLS has been a long time ago. Why is it applicable to this research object? The same is true for other scales.

4.      To compare donor and control, it must be verified that there is no difference in the basic information of the two groups.  Otherwise, it is not meaningful to compare.

5.      The depth and breadth of discussions should be strengthened.

Author Response

Dear Reviewer,

Response to Reviewer 3 Comments

Thank you for your very careful review of this manuscripts, and for the suggestions that ensued. The responses are given as below:

Point 1: The cited literature is too old, it is recommended to add some recent

literature.

Response 1: Thanks for the suggestion. After carefully reviewing the latest research findings, the researcher integrated them into the literature review and discussion.

Recent studies in the reference

  1. Orsini, E.; Quaranta, M.; Ratti, S; Mariani, G.A.; Mongiorgi, S.; Billi, A.M.; Manzoli, The whole body donation program at the university of Bologna: A report based on the experience of one of the oldest university in Western world. Ann Anat. 2021 ,234:151660. doi: 10.1016/j.aanat.2020.151660.
  2. Chinni, M.; Hubley, A. M. A research synthesis of validation practices used to evaluate the

Satisfaction with Life Scale (SWLS). In Validity and Validation in Social, Behavioral, and Health Sciences; Zumbo, B.D.; Chan, E.K.H., Eds.; Springer, NY, 2014; pp. 35–67.

  1. Smith, C.F.; Munro. R.; Davies, D.C.; Wilkinson, T.; Shaw, H.; Claridge, K.; Llewellyn, S.; Mc Ateer, P.; Ward, S.; Farsides, T. Understanding beliefs, preferences and actions amongst potential body donors. Anat Sci Educ. 2023, 16(2):224-236. doi: 10.1002/ase.2204.
  2. Jenkin, R.A.; Garrett, S.A.; Keay, K.A. Altruism in death: Attitudes to body and organ donation in Australian students. Anat Sci Educ. 2023, 16(1):27-46. doi: 10.1002/ase.2180
  3. Mueller, C.M.; Allison, S.M.; Conway, M.L. Mississippi's whole body donors: Analysis of donor pool demographics and their rationale for donation. Ann Anat. 2021, 234:151673. doi: 10.1016/j.aanat.2020.151673
  4. Farsides, T.; Smith, C. F.; Sparks, P. Beyond “altruism motivates body donation”, Death Stud. 2023, 47:1, 56-64, doi: 10.1080/07481187.2021.2006827

Point 2: The research method should be explained in more detail. Why choose to use convenience sampling instead of all the volunteers in Jing Si Halls as the sample group?

Response 2: Thanks for the comments. Due to the sheer number of volunteers of Tzu Chi foundation, which exceeds ten thousand, it is impractical to enroll all the volunteers in Jing Sing Halls into our study.

We invited individuals who are interested in participating in the study to fill out the questionnaires.

Point 3: However, the research subjects come from two completely different living areas in northern Taiwan and eastern Taiwan. Whether the region will cause differences should be included in the control.

Response 3: Thanks for the comments. A group comparison analysis was carried out to examine whether significant regional difference (northern versus eastern) exists on the research variables.   

An independent t-test was performed to test whether significant regional difference (northern versus eastern Taiwan) exists on the research variables. The results revealed no significant differences on scores of  death anxiety (t(679) = -.25, p = .81), gratitude (t(679) = 1.15, p = .25), altruism (t(679) = .32, p = .75), life meaning (t(679) = .40, p = .69), and life satisfaction (t(679) = 1.04, p = .30). (p. 4)

Point 4: Reasons should be given for choosing the scale to be used. For example, there are many quality of life scales. Why choose the SWLS developed in 1985? The development of SWLS has been a long time ago. Why is it applicable to this research object? The same is true for other scales.

Response 4: Thanks for pointing this out. We addressed this issue in the revised manuscript.  

Despite consisting of only five items, the SWLS has consistently demonstrated good psychometric properties. This has contributed to its widespread use in research and clinical settings as a reliable and valid tool for assessing an individual's subjective well-being [30]. (p.3)

Point 5: To compare donor and control, it must be verified that there is no difference in the basic information of the two groups.  Otherwise, it is not meaningful to compare.

Response 5: Thanks for the thoughtful comments. We have conducted an analysis to confirm that there are no significant differences between the two groups in terms of demographic variables.

 Prior to examining the factors influencing the decision to body donation, a series of chi-square tests were performed to compare demographic variables of the registered-donor and the control groups. No significant differences were observed between the two groups in terms of demographic characteristics, including gender (χ2 (df = 1) = 0.06, p = .81), level of education (χ2 (df = 4) = 1.70, p = .79), employment status (χ2 (df = 1) = 1.31, p = .25), and marital status (χ2 (df = 2) =0.99, p = .61). (p.4)

Point 6: The depth and breadth of discussions should be strengthened.

Response 6: Thanks for pointing this out. Several paragraphs of discussion were rewritten.

The aims of the present study was to investigate the psychological characteristics and motivations of individuals who have registered as body donors for medical science in Taiwan. The results of the study revealed that the primary motivation for body donation was the desire to be useful to society after death and to advance medical education and research. Interestingly, the study also found that although women were more likely to pledge to donate their bodies, the majority of donated bodies were male. The study further identified the key factors that predict pledging to donate one's body to medical science. The results showed that altruism was the most important factor in predicting donation, and that registered donors exhibited higher levels of altruism compared to the control group. Additionally, donors had lower levels of anxiety related to death and stronger feelings of gratitude, purpose in life, and life satisfaction compared to non-donors. However, the study found that only death anxiety and gratitude significantly predicted an individual's likelihood of donating their body to medical science. (p. 6)

Most of the donors surveyed in the present study indicated that they wanted to help advance medical education and research, which is in line with previous research [2, 39, 42-48]. A remarkable consistency in motivational factors for body donation has been observed across various studies worldwide, which includes desires to contribute to medical education or scientific research, or be useful or helpful. Farsides et al., [45] classified these motivations as “medical altruism”, referring to individuals who seek to benefit healthcare professionals or institutions. In addition, the desire to spare one’s family the trouble and expense related to one’s funeral was one of the main motivations among the male participants in this study, but not the female participants. Farsides et al.,[45] labeled the motivation as “intimate altruism”, where some individuals choose to forego a traditional funeral, at least in part, with the aim of preventing their loved ones from bearing the cost, grief, or inconvenience associated with it. In Chinese societies such as Taiwan, male relatives are expected to play a more prominent role in the funerals of family members [5]; therefore, it is likely that their past experiences made male donors more aware of and sensitive to issues related to arranging funerals and interment or cremation. (p.7)

Donors and non-donors in the present study demonstrated significant differences in the measures of gratitude and life satisfaction. However, only a high sense of gratitude was found to be a strong predictor of donating one’s body to medical science. Similarly, McCullough et al. [61] found that a relatively high sense of gratitude, whether self- or other-rated, was a reliable predictor of prosocial behaviors. The sense of gratitude is a type of ethical sentiment associated with concern for the well-being of others, predisposing one to altruistic behavior [62]. Bolt et al. [63] found that a large proportion of people who donated their bodies to medical science were motivated by the desire to express gratitude for medical care received in the past or the benefits brought to society in general by advancements in medical science. Some registered donors may express gratitude for the positive impact that medical science has had on the health and well-being of their family. When a person receives successful treatment, their family ofen feel a sense of relief and gratitude as well. This shared gratitude can inspire a desire to give back to medical science in some way. (p.8) 

For more details please see the revised version manuscript. 

Round 2

Reviewer 3 Report

Complete supplements and statements have been made for the last review opinion, which shows the author's intentions.

Agree to publish.